# Fokker–Planck Analysis of Superresolution Microscopy Images

**Mario Annunziato [1],\* and Alfio Borzì [2]**

[1]  Dipartimento di Fisica "E. R. Caianiello", Università degli Studi di Salerno, Via G. Paolo II 132, 84084 Fisciano, Italy

[2]  Institut für Mathematik, Universität Würzburg, Emil-Fischer-Strasse 30, 97074 Würzburg, Germany; alfio.borzi@mathematik.uni-wuerzburg.de

\*  Correspondence: mannunzi@unisa.it or mannunzi@am-research.it

**Abstract:** A method for the analysis of super-resolution microscopy images is presented. This method is based on the analysis of stochastic trajectories of particles moving on the membrane of a cell with the assumption that this motion is determined by the properties of this membrane. Thus, the purpose of this method is to recover the structural properties of the membrane by solving an inverse problem governed by the Fokker–Planck equation related to the stochastic trajectories. Results of numerical experiments demonstrate the ability of the proposed method to reconstruct the potential of a cell membrane by using synthetic data similar those captured by super-resolution microscopy of luminescent activated proteins.

**Keywords:** super-resolution microscopy; Fokker–Planck equation; stochastic processes; numerical optimization





## 1. Introduction

The pioneering works [1–3] mark the development of the revolutionary superresolution microscopy (SRM) that allows us to go beyond the Abbe limit for conventional light microscopy [4]. The SRM method consists of labeling the molecules moving on a biological support with fluorophores and then in sampling the microscopic images of the activated fluorescent molecules.

Observation of the frames of the sampled SRM microscopic images have suggested that the motion of the molecules could be modeled by a stochastic Langevin equation [5,6]. Clearly, the cell membrane at the microscope is a 3-dimensional object; however, it can be considered flat, and the observed motion is 2-dimensional since it results in the projection on the focal plane of the SRM microscope. It appears that an adequate model of the observed trajectories of 2-dimensional images is given by the following stochastic differential equation (SDE) [7]:

$$dX_t = b(X_t)\, dt + \sigma(X_t)\, dW_t \tag{1}$$
$$X_{t_0} = X_0, \tag{2}$$

where $b$ represents the drift, $\sigma$ is the dispersion coefficient, and $X_t \in \mathbb{R}^2$ denotes the position of the observed molecule at time $t$. In this framework, it is well-known that the drift and dispersion coefficients satisfy

$$\lim_{t \to s} \mathbb{E}\left[ \frac{X_t - X_s}{t - s} \mid X_s = z \right] = b(z), \qquad \lim_{t \to s} \mathbb{E}\left[ \frac{|X_t - X_s|^2}{t - s} \mid X_s = z \right] = \sigma^2(z),$$

where the expected values are computed with respect to the process having value $z$ at $t = s$; the operator $\mathbb{E}[\cdot \mid X_s = z]$ denotes averaging with regard to the measure of the trajectories conditioned to be at $z$ at time $s$.

The formulas above suggest that suitable approximations to $b$ and $\sigma$ can be obtained by tracking single molecules; see, e.g., [7,8]. However, this approach may suffer from the highly fluctuating values of the trajectories and the difficulty of discerning between different molecules that come closer to the resolution limit.

For this reason, already in [9] the authors have pursued an alternative strategy that allows us to build a robust methodology for the estimation of the drift based on the observation of an ensemble of trajectories. Our approach is built upon the assumption that this ensemble is driven by a velocity field (the drift), given by a potential velocity field $U(x)$, with $x \in \Omega \subset \mathbb{R}^2$, as follows:

$$b(x; U) = -\nabla U(x). \tag{3}$$

Moreover, one assumes a constant diffusion coefficient whose value is chosen consistently with estimates of laboratory measurement [10]. It is the purpose of our work to reconstruct the potential $U(x)$ by means of the observation of the motion $X_t$ of the molecules modeled by Equation (1).

In agreement with our statistical approach based on ensembles, we focus on the evolution of the probability density function (PDF) of the positions of the molecules (not on the single trajectories) whose evolution is governed by the Fokker–Planck (FP) problem given by [6,11]:

$$\partial_t f(x, t) - \nabla \cdot (\nabla U(x) f(x, t)) - \frac{\sigma^2}{2} \Delta f(x, t) = 0, \quad (x, t) \in Q \tag{4}$$

$$F(f) \cdot \hat{n} = 0, \quad (x, t) \in \Sigma, \tag{5}$$

$$f(x, 0) = f_0(x), \quad x \in \Omega, \tag{6}$$

where $Q = \Omega \times (0, T)$ and $\Sigma = \partial\Omega \times (0, T)$. In this formulation, $f(x, t)$ represents the PDF of a particle at $x \in \mathbb{R}^2$ at time $t$, $\nabla U(x)$ is the Cartesian gradient of the potential $U$, $f_0$ is the initial density, and $\Delta$ is the two-dimensional Laplace operator. Notice that we require zero-flux boundary conditions, where $F(f)(x, t)$ is the following flux of probability

$$F(f)(x, t) = \frac{\sigma^2}{2} \nabla f(x, t) - b(x; U) f(x, t). \tag{7}$$

We choose zero-flux boundary conditions since they reasonably model the situation where a similar number of particles enters and exits the domain; see, e.g., [6,12].

Our proposal is to construct an FP-based imaging modality that is based on the formulation of an inverse problem for $U$ and the observation of a time sequence, in time interval $[0, T]$, of numerical PDFs (two-dimensional histograms), which are obtained from a uniform binning of SRM particles' positions. We denote this input data as $f_d(x, t)$ which is a piecewise constant function. In this setting, the initial condition is given by $f_0(x) = f_d(x, 0)$.

This proposal is similar to that in our previous work [9]. However, in [9] the assumption of interacting particles was made that resulted in very involved and CPU time demanding calculations. It is the purpose of this work to demonstrate that accurate reconstruction results can be obtained assuming noninteracting particles, hence by using a linear FP model.

At the continuous level, our FP-based imaging tool is formulated as the following inverse problem:

$$\min J(f, U) := \frac{1}{2} \int_\Omega \int_0^T (f(x,t) - f_d(x,t))^2 \, dx \, dt$$

$$+ \frac{\xi}{2} \int_\Omega (f(x,T) - f_d(x,T))^2 \, dx$$

$$+ \frac{\alpha}{2} \int_\Omega (|U(x)|^2 + |\nabla U(x)|^2) \, dx, \qquad (8)$$

$$s.t. \quad \partial_t f(x,t) + \nabla \cdot [b(x,;U)f(x,t)] - \frac{\sigma^2}{2} \Delta f(x,t) = 0, \quad \text{in } Q$$

$$f(x,0) = f_0(x) \text{ in } \Omega, \qquad F(f) \cdot \hat{n} = 0 \text{ on } \Sigma,$$

with the given initial and boundary conditions for the FP equation, and $\alpha, \xi > 0$.

In this problem, the objective functional $J$ is defined as the weighted sum of a space–time best fit term $\int_\Omega \int_0^T (f(x,t) - f_d(x,t))^2 \, dx \, dt$, and at final time $\int_\Omega (f(x,T) - f_d(x,T))^2 \, dx$, and of a suitable 'energy' of the potential $\|U\|^2 = \int_\Omega (|U(x)|^2 + |\nabla U(x)|^2) \, dx$, which corresponds to the square of the $H^1(\Omega)$ norm of $U$. Notice that this formulation allows us to avoid any differentiation of the data and makes it possible to choose the binning size and, in general, the measurement setting, independently of any choice of parameters that are required in the numerical solution of the optimization problem.

Our second main concern in determining the potential $U$ is to provide a measure of uncertainty, and thus of reliability, of its reconstruction. Statistically, this is achieved by many repetitions of the same experiment, that could not be feasible for (short) living cells. However, inspired by the so-called model predictive control (MPC) scheme [13] already used for optimal control problems [12,14], we propose a novel procedure to quantify the uncertainty of the estimation of $U$ by using the data of a single experiment.

Our methodology is to consider a sequence of non-parametric inverse problems like (8) defined on time windows $(t_k, t_{k+1})$, $k = 0, \dots, K - 1$, that represent a uniform partition in $K$ subintervals of the time interval $[0, T]$. Therefore, a statistical analysis can be performed on the set of the corresponding $K$ solutions for $U$ that are obtained in the subintervals.

For development and validation, we consider images of a cell's membrane structures (actin, cytoskeleton), as expression of potentials, that is pixel grey values where increased brightness stands for more repulsion, with which we generate our synthetic data. In particular, we use an image of actin from a cytoskeleton obtained with a Platinium-replica electron microscopy [15,16].

With this images taken as gray-level representation of potential functions, we perform Monte Carlo simulation of motion of particles to generate images of molecules at different time instants, thus constructing the datasets representing the output of measurements. This setting is illustrated in Figure 1, where the image of actin [17] and a plot of few trajectories of the corresponding stochastic motion of the particles in this potential are shown.

Once the synthetic measurement data is constructed, we perform a pre-processing step on this data to construct the numerical PDF required in our method and solve our inverse problem to find the estimated–measured potential $U$. The latter is compared with that one used in the MC simulation, by a measure of similarity based on the pixel cross-correlation between the two images.

In Section 2, we discuss a numerical methodology for solving our FP-based reconstruction method for the potential $U$. In Section 3, we provide all details of our experimental setting and introduce some analysis tools for determining the accuracy of the proposed reconstruction. In Section 4, we validate our reconstruction method, and use our uncertainty quantification procedure. In Section 5, we investigate the resolution of the proposed FP image reconstruction as an optical instrument. A section of conclusion and acknowledgements completes this work.

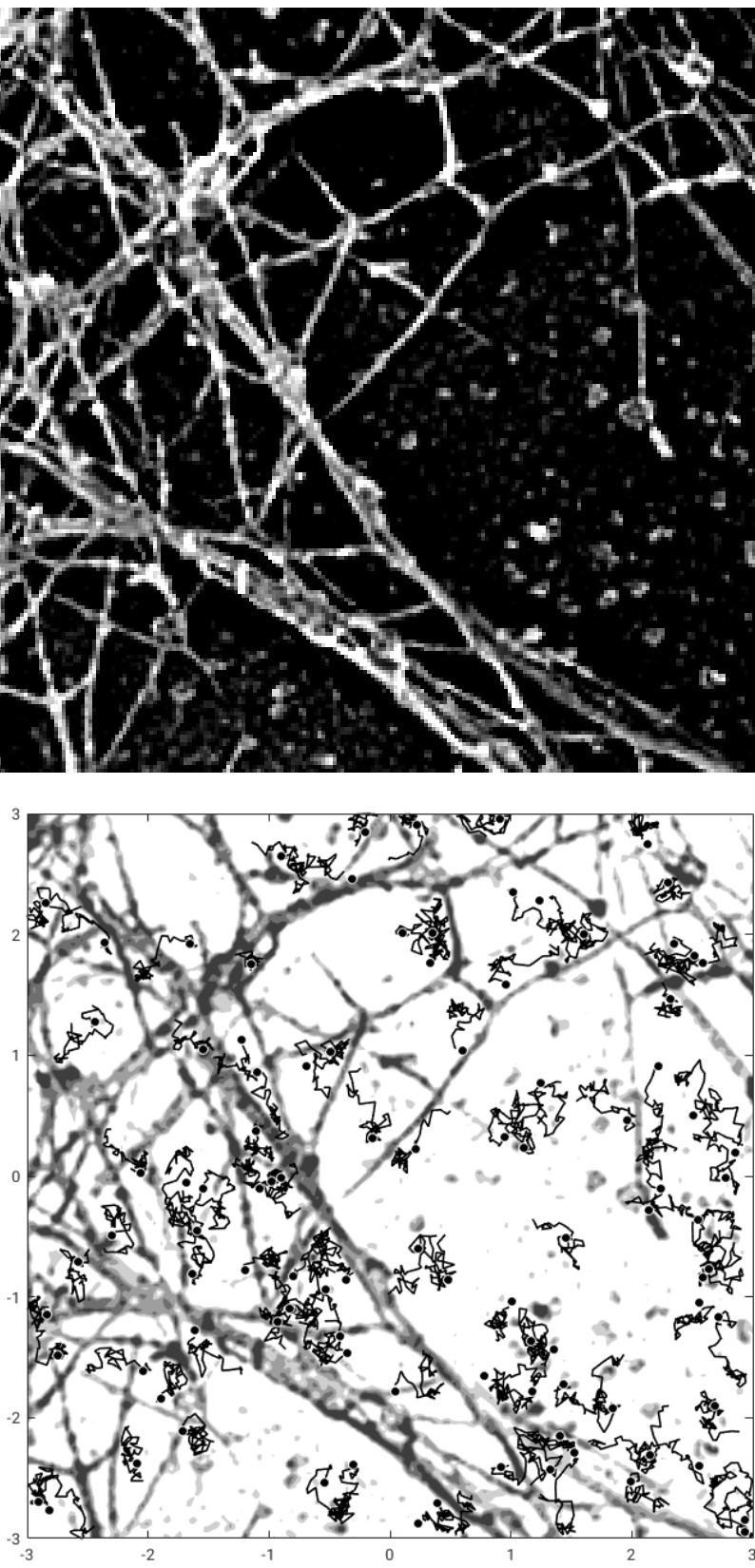

**Figure 1.** (**Up**) A picture of actin from a cytoskeleton as cell membrane potential (close up) (courtesy of Koch Institute [17]); (**Down**) a few simulated trajectories of particles (black dots) on the membrane (in reverse colors).

## 2. Numerical Methodology

Our aim is to reconstruct the potential $U$ from the data consisting of a temporally sampled SRM images of the positions of particles subject to this potential; see Figure 2 (up) for a schematic snap-shot of this data. This image is subject to a pre-processing binning procedure in order to construct histograms by counting the number of particles in a regular square partition of $\Omega$. The height of an histogram is proportional to the number of particles in a bin of the domain. This procedure for the image at time $t$ defines the histogram function $f_d(\cdot, t)$; see Figure 2 (down).

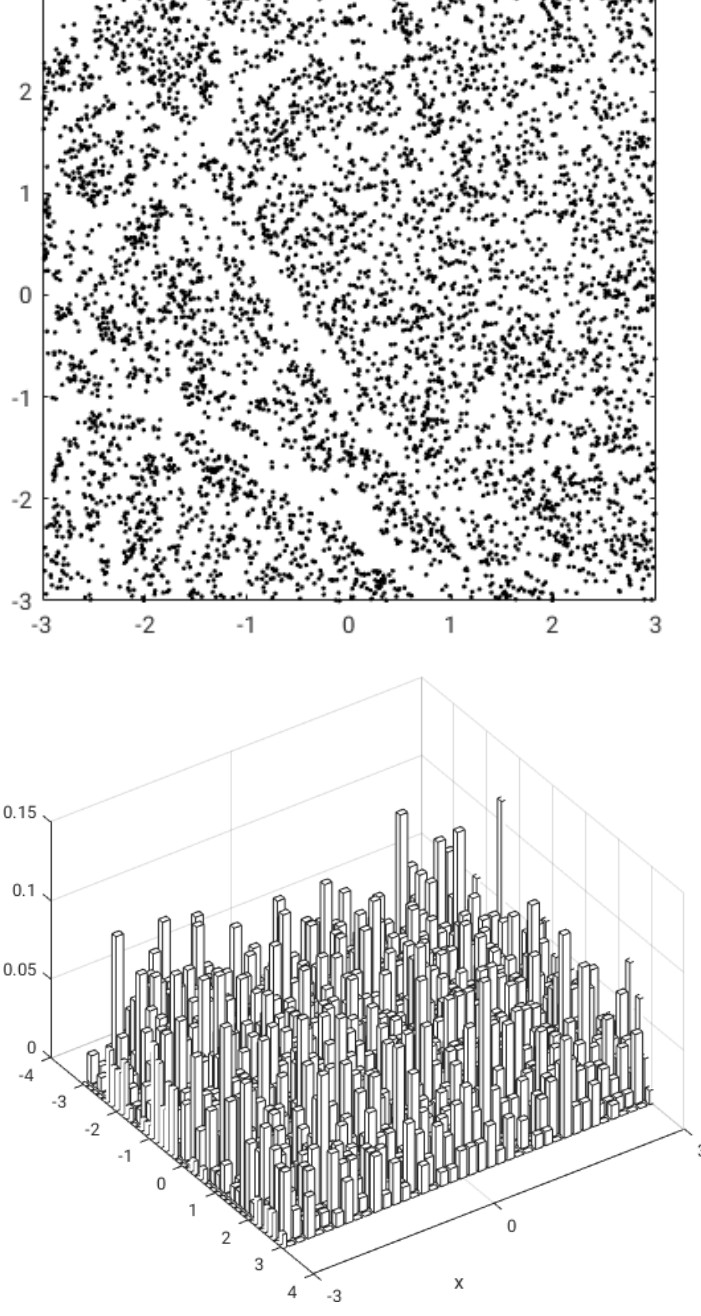

**Figure 2.** A frame of particles (**up**) and the corresponding histogram $f_d(x, t)$ on a mesh of $40 \times 40$ bins for a fixed time (**down**), from simulated data.

In order to illustrate our numerical framework, we introduce the potential-to-state map $U \mapsto f = S(U)$, that is, the map that associates to a given $U \in H^1(\Omega)$ the unique

solution to our FP problem (4)–(6), with given initial condition $f_0$. For the analysis of well-posedness and regularity of the map $S$, we refer to [18].

Next, we remark that with the map $S$, we can define the reduced objective functional $\hat{J}(U) := J(S(U), U)$ and consider the equivalent formulation of (8) given by

$$\min_{U \in H^1(\Omega)} \hat{J}(U),\tag{9}$$

which has the structure of an unconstrained optimization problem. Thanks to the regularity of $S$ and the quadratic structure of $J$, existence of an optimal $U$ can be stated by well known techniques; see, e.g., [19].

Further, since $S$ and $J$ are Fréchet differentiable, it is possible to characterize an optimal $U$ as the solution to the following first-order optimality condition

$$\nabla_U \hat{J}(U) = 0,$$

where $\nabla_U \hat{J}(U)$ denotes the so-called reduced gradient [20].

In the Lagrange framework, this condition results in the following optimality system:

$$
\begin{aligned}
&\partial_t f(x,t) + \nabla \cdot [b(x;U) f(x,t)] - \frac{\sigma^2}{2} \Delta f(x,t) = 0, \\
&f(x,0) = f_0(x) \text{ in } \Omega, \qquad F(f) \cdot \hat{n} = 0 \text{ on } \partial\Omega \times (0,T], \\
&\partial_t p(x,t) + \frac{\sigma^2}{2} \Delta p(x,t) + \nabla p(x,t) \cdot b(x;U) = f(x,t) - f_d(x,t), \\
&p(x,T) = -\xi \left( f(x,T) - f_d(x,T) \right) \text{ in } \Omega, \qquad \partial_{\hat{n}} p(x,t) = 0 \text{ on } \partial\Omega \times (0,T], \\
&\alpha\, U(x) - \alpha\, \Delta U(x) - \int_0^T \nabla \cdot (f(x,t)\, \nabla p(x,t)) dt = 0 \text{ in } \Omega, \\
&\qquad \partial_{\hat{n}} U = 0, \qquad \text{on } \partial\Omega,
\end{aligned}
\tag{10}
$$

where $p$ denotes the adjoint variable, which is governed by a backward adjoint FP equation. Notice that the adjoint equation is a well-posed problem with $f_d(x,t) \in L^2$, which allows to use the irregular histograms as input data. The numerical solution obtained with a finite difference approximation is consistent with the interpretation of $f_d$ as a piecewise constant function obtained by local averaging an $L^2$ function on subcells centered at grid points. Notice that the numerical grid is finer than that of the binning.

The last equation in (10) is the so-called the optimality condition equation, and the Neumann boundary condition $\partial_{\hat{n}} U = 0$ is our modeling choice. One can show that its left-hand side represents the $L^2$ gradient along the FP differential constraint with respect to $U$ of the objective functional. We have

$$\nabla_U \hat{J}(U)(x) := \alpha\, U(x) - \alpha\, \Delta U(x) - \int_0^T \nabla \cdot (f(x,t)\, \nabla p(x,t)) dt.\tag{11}$$

Our approach for solving our FP optimization problem (8) is based on the nonlinear conjugate gradient (NCG) method; see, e.g., [20]. This is an iterative method that resembles the standard CG scheme and requires to estimate the reduced gradient $\nabla_U \hat{J}(U)$ at each iteration.

In order to illustrate the NCG method, we start with a discussion on the construction of the gradient. For a given $U^n$ obtained after $n$ iterations, we solve the FP equation and its adjoint, and use (11) to assemble the $L^2$ gradient. However, since the potential is sought in $H^1(\Omega)$, we need to obtain the $H^1$ gradient that satisfies the following relation

$$\left( \nabla_U \hat{J}(U)|_{H^1}, \delta U \right)_{H^1} = \left( \nabla_U \hat{J}(U)|_{L^2}, \delta U \right)_{L^2},\tag{12}$$

where $(\cdot, \cdot)$ denotes the $L^2(\Omega)$ scalar product.

Now, using the definition of the $H^1$ inner product, we obtain

$$\int_\Omega \left[ \nabla_U \hat{J}(U)|_{H^1} \cdot \delta U(x) + \nabla_x \nabla_U \hat{J}(U)|_{H^1} \cdot \nabla_x \delta U(x) \right] dx = \int_\Omega \nabla_U \hat{J}(U)|_{L^2} \, \delta U(x) \, dx, \quad (13)$$

which must hold for all the test functions $\delta U \in H^1(\Omega)$. Therefore we obtain

$$-\Delta \left[ \nabla_U \hat{J}(U)|_{H^1} \right] + \left[ \nabla_U \hat{J}(U)|_{H^1} \right] = \nabla_U \hat{J}(U)|_{L^2}, \quad (14)$$

with the boundary conditions $\frac{\partial}{\partial \hat{n}} \nabla_U \hat{J}(U)|_{H^1} = 0$ on $\partial \Omega$; see [9] for more details. In Algorithm 1 our procedure for computing the gradient is given.

---

**Algorithm 1** Calculate $H^1$ gradient.

---

**Require:** control $U(x)$, $f_0(x)$, $f_d(x,t)$.
**Ensure:** reduced gradient $\nabla_U \hat{J}(U)|_{H^1}$
   Solve forward the FP equation with inputs: $f_0(x)$, $U(x)$
   Solve backward the adoint FP equation with inputs: $U(x)$, $f(x,t)$
   Assemble the $L^2$ gradient $\nabla_U \hat{J}(U)|_{L^2}$ using (11).
   Compute the $H^1$ gradient $\nabla_U \hat{J}(U)|_{H^1}$ solving (14).
   **return** $\nabla_U \hat{J}(U)|_{H^1}(x)$

---

In this algorithm, the FP problem and its optimization FP adjoint are approximated by the exponential Chang–Cooper scheme and the implicit BDF2 method; see [12].

Now, we can discuss the NCG method. The NCG iterative procedure is initialized with $U^0(x) = 0$. We denote the optimization directions with $d^n$. In the first update, we have $d^0 = -\nabla_U \hat{J}(U^0)|_{H^1}$ and perform the optimization step

$$U^1 = U^0 + \alpha_0 \, d^0,$$

where $\alpha_0$ is obtained by a backtracking linesearch procedure. After the first step, in the NCG method the descent direction is defined as a linear combination of the new gradient and the past direction as follows:

$$d^n = -\nabla_U \hat{J}(U^n)|_{H^1} + \beta_{n-1} \, d^{n-1},$$

where $\beta_{-1} = 0$, and $\beta_{n-1} = \|\nabla_U \hat{J}(U^n)|_{H^1}\|^2 / (d^{n-1} \cdot (\nabla_U \hat{J}(U^n)|_{H^1} - \nabla_U \hat{J}(U^{n-1})|_{H^1}))$, that is, the Dai-Yuan formula $b_{n-1} = \|r_n\|^2 / (-d_{n-1}(r_n - r_{n-1}))$, where here $r_n$ stand for the deepest descent direction $r_n = -\nabla_U \hat{J}(U^n)|_{H^1}$ and $d_{n-1}$ is the conjugate direction at the previous step; see, e.g., [20].

The tolerance *tol* and the maximum number of iterations $n_{\max}$ are used for termination criteria. Summarizing, in Algorithm 2 we present the NCG procedure.

---

**Algorithm 2** Nonlinear conjugate gradient (NCG) method

---

**Require:** $U^0(x) \equiv 0$, $f_0(x)$, $f_d(x,t)$
**Ensure:** Optimal control $U(x)$ and corresponding state $f(x,t)$
   $n = 0$
   Assemble gradient $g^0 = \nabla_U \hat{J}(U^0)|_{H^1}$ using Algorithm 1; set $d^0 = -g^0$.
   **while** $\|g^n\|_{H^1} > tol$ **and** $n < n_{\max}$ **do**
      Use linesearch to determine $\alpha_n$
      Update control: $U^{n+1} = U^n + \alpha_n \, d^n$
      Compute the gradient $g^{n+1} = \nabla_U \hat{J}(U^{n+1})|_{H^1}$ using Algorithm 1
      Calculate the new descent direction $d^{n+1} = -g^{n+1} + \beta_n \, d^n$
      Set $n = n + 1$
   **end while**
   **return** $U^n(x)$

---

For our numerical experiments these algorithms have been implemented with object oriented programming in C++, by using the numerical libraries Armadillo [21,22], Openblas [23], Lapack [24], SuperLU [25,26] and HDF5 [27].

## 3. Experimental Design and Analysis Tools

In a real application, the input data $f_d$ is given by frames of a recorded sequence of a SRM experiment, and the desired output is the reconstructed potential denoted with $U_r$. In our case, we construct this data based on a sample potential $U_s$ that determines the drift function in our stochastic model (1). Thus, we generate our frames of synthetic data first by time-integrating this SDE in the chosen interval $[0, T]$, and choosing the initial positions of the particles randomly uniformly distributed. Next, the positions of the particles at different times are collected in a sequence of 2-dimensional bins that result in the sequence of distributions $f_d(x, t_\ell)$, $\ell = 1, \dots, L$, where $L$ is the length of the resulting time sequence of frames. With this preparation, we apply our algorithm to obtain $U_r$, which represents the proposed reconstruction of $U_s$. A comparison between these two potentials allows to validate the accuracy of our reconstruction method (see below).

We choose a domain $\Omega = [-3, 3] \times [-3, 3]$, and $U_s$ corresponds to Figure 1 (up), where the values of $U_s$ in $\Omega$ correspond to the gray scale pixel values of the picture mapped in $[0, 1]$. With this $U_s$, we perform a stochastic simulation of $N_p$ particles for a time horizon $T$, and diffusion amplitude $\sigma$. The particles trajectories given by (1) with (3), are computed with the Euler-Maruyama scheme with a time step $\tau = 10^{-3}$, which results in a number of $L$ frames. In this simulation, reflecting barriers for the stochastic motion are implemented. We remark that for the following calculations we are going to consider a relatively small value of density of particles; see [10,28].

Next, we perform a binning of the positions of the particles at each frame to construct $f_d$. Hence, we consider a uniform partition of $\Omega$ with non-overlapping squares; see Figure 2 for a plot of particles in $\Omega$ at a given time and the corresponding $f_d$. Notice that $f_d$ is irregular; nevertheless, we do not perform smoothing of this data. The sequence of $f_d$ values enter in our best fit functional in (8).

Once we have computed $U_r$ with our optimization procedure, we aim at providing a quantification of its uncertainty. Thus, we compute the following normalized cross-correlation factor between the reconstructed potential $U_r$ and the one used to generate the synthetic data $U_s$. We have

$$cc(U_r, U_s) = \frac{U_r \cdot U_s}{|U_r| \, |U_s|}. \tag{15}$$

In this formula, $U_r$ and $U_s$ are considered as vectors and $\cdot$ represents the scalar product. Therefore if $cc = 1$ we have that the two potentials match perfectly, whereas if its value is close to 0, the two potentials are dissimilar. Notice that cross-correlation is commonly used in medical imaging and biology; see, e.g., [29–31].

Clearly, one could consider many repetitions of the simulation of the motion of the particles with the same initial condition and make the final binning on the average of the resulting frames. This procedure would result in a less fluctuating $f_d(x, t_\ell)$ that allows a better reconstruction. However, this scenario seems difficult to realize in the real laboratory setting of a living cell. On the other hand, in SRM, imaging is able to visualize the motion of the particles on a cell membrane for a relatively long time ($T \gg 1$ in our setting), and our approach exploits this possibility considering a subdivision of the time interval in a number $K$ of time windows, and solving our optimization problem in each of these windows almost independently. This approach allows us to improve the reconstruction $U_r$ and makes it possible to quantify the uncertainty of the reconstruction.

Now, to illustrate our approach, consider a uniform partition of $[0, T]$ in time windows of size $\Delta t = T/K$ with $K$ a positive integer. Let $t_k = k\Delta t$, $k = 0, 1, \dots, K$, denote the start- and end-points of the windows. At time $t_0$, we have the initial PDF $f_0$, and we solve our optimization problem (8) in the interval $[t_0, t_1]$. This means that the final time is $t_1$ and the terminal condition for the adjoined variable is given by $p(x, t_1) = -\xi \left( f(x, t_1) - f_d(x, t_1) \right)$.

The resulting potential is denoted with $U_1$. Thus, the solution obtained in this window also provides the PDF at $t = t_1$.

Clearly, we can repeat this procedure in the interval $(t_1, t_2)$ with the computed PDF at $t = t_1$ as the initial condition and $t_2$ as final time, to compute $U_2$. This procedure is recursive and can be repeated for $k = 1, \ldots, K$, thus obtaining $U_k$, $k = 1, \ldots, K$.

Notice that small values of $K$ in relation to $L$ produce a rough estimate of the average potential and its standard deviation due to statistical fluctuations of the Monte Carlo experiments. On the other hand, for greater values of $K$, the number of frames for each window of our approach is reduced when $L$ is kept fixed, thus resulting in a worsening of the reconstruction procedure.

For the purpose of our analysis, we apply a scaling of these potentials so that their point-wise values are in the interval $[0, 1]$. This scaling is performed as follows:

$$\hat{U} = \frac{U - \min(U)}{\max(U) - \min(U)}. \tag{16}$$

Thereafter, we the reconstructed potential by pixel-wise average of the $U_k$ is given by

$$\langle U_r \rangle = \frac{1}{K} \sum_{k=1}^{K} \hat{U}_k. \tag{17}$$

Moreover, we can also compute the following pixel-wise standard deviation

$$sd(U_r) = \sqrt{\sum_{k=1}^{K} \frac{(\hat{U}_k - \langle U_r \rangle)^2}{K - 1}}. \tag{18}$$

Next, we provide conversion formulas for our parameters in order to accommodate data from real laboratory experiments. We introduce a unit of length u such that the side length $l$ of our square domain $\Omega$ is given by $l = 6$ u, and the unit of the noise amplitude $\sigma$ is given by $\sqrt{u}/s$. In real biological experiments, the typical measure of the length $\tilde{l}$ of a cell membrane is given in μm. Further, the particle's diffusion constant $D = \sigma^2/2$ is given in μm$^2$/s; hence, we have the correspondence $\sigma = l/\tilde{l}\sqrt{2D}$ in unit $\sqrt{u}/s$, whose value is used for MC simulations.

The depth of the potential $\tilde{U}$ is expressed in unit of $K_B\bar{T}$, where $K_B$ is the Boltzmann constant and $\bar{T}$ the absolute temperature. In experimental papers, the Equation (3) is written with the diffusion constant $D$ and $K_B\bar{T}$, i.e., $D\tilde{U}/(K_B\bar{T})$. As above, we obtain the relationship between the values of a potential $U$ and the scaled $\tilde{U}$, as $\tilde{U} = U(\tilde{l}/l)^2/D$ in the unit of $K_B\bar{T}$.

As an illustration of the setting above, we see that in an experiment, the super-resolution of an acquired image frame can reach the value of 0.02 μm/pixel. With an image of $500 \times 500$ pixels, we have $\tilde{l} = 10$ μm. The average diffusion coefficient of particles (protein molecules) observed in SRM imaging is estimated with $D = 0.1$ μm$^2$/s [10]. By super-resolution techniques, it is possible to activate a density of $0.5 \div 2/\text{μm}^2$ visible particles, which in terms of image pixels corresponds to $0.5 \div 2$ particles in a square of 50 pixels of side. Each frame is usually sampled at time intervals of $\delta t = 30$ ms.

In order to set up a consistent MC simulation of a real experiment, by mapping an image of a square of side 10 μm on our domain $\Omega$, we get from the above mentioned formula: $\sigma = 6/10\sqrt{0.2} \approx 0.268$ u$/\sqrt{s}$.

## 4. Numerical Validation

In this section, we discuss results of experiments in a setting that is close to real laboratory experiments involving SRM imaging. The results of these experiments demonstrate the ability of our methodology to reconstruct the potential from the simulations of the SRM measurements of the motion of particles on a cell's membrane.

We consider a potential that corresponds to a portion of cytoskeleton as depicted in Figure 3, with $200 \times 200$ pixels. We assume that the pixel is 50 nm, which corresponds to an area of $100 \, \mu m^2$. In the figure, the white regions represent the structure of the cytoskeleton; the black ones are the 'valleys' where the proteins are supposed to be attracted.

For the MC simulations for generating the synthetic data, we choose $\sigma = 0.268 \, u/\sqrt{s}$. This value of $\sigma$ corresponds to a diffusion constant of $D \simeq 0.1 \, \mu m^2/s$. We consider $N_p = 1000$ particles, i.e., an average density of 10 particles per $\mu m^2$. In this case, we consider a sequence of $L = 3000$ frames and $T = 90$, obtained by the numerical integration of the stochastic differential equation with an integration step $\tau = 30 \times 10^{-3}$ s. The frames have $\delta t = 30$ ms, similar to a real experiment. The resulting (single run) particle trajectories are collected in a binning process based on a mesh $\Omega$ of $50 \times 50$ bins.

For our reconstruction method, we choose a numerical partition of $\Omega$ of $100 \times 100$ subdivisions, corresponding to a mesh size of 100 nm. The time integration step coincides with that of the frames. For the tracking functional, we set $\alpha = 10^{-4}$ and $\xi = 1$. Further, in the FP setting, we have $\sigma = 0.7 \, u/\sqrt{s}$. Notice that $\sigma$ in the FP model is chosen to be larger than the one used in the MC simulations. This choice is dictated by numerical convenience and it appears that it does not affect the quality of the reconstruction. The calculations are performed according to the MPC procedure with $K = 5$ time windows.

With this setting, we obtain the reconstructed potential shown in Figure 3 (down). We see that the reconstruction is less sharp as we expected considering the much finer structure of the cytoskeleton and the small number of particles involved.

Further, in Figure 4, we depict the potentials obtained on each time window of the MPC procedure and the values of the corresponding $cc$. With these results, we have obtained the reconstructed potential $U_r$ in Figure 3, which we re-plot in level-set format in Figure 5 for comparison. In Figure 5, we also depict the standard deviation that suggests that we have obtained a reliable reconstruction with small uncertainty.

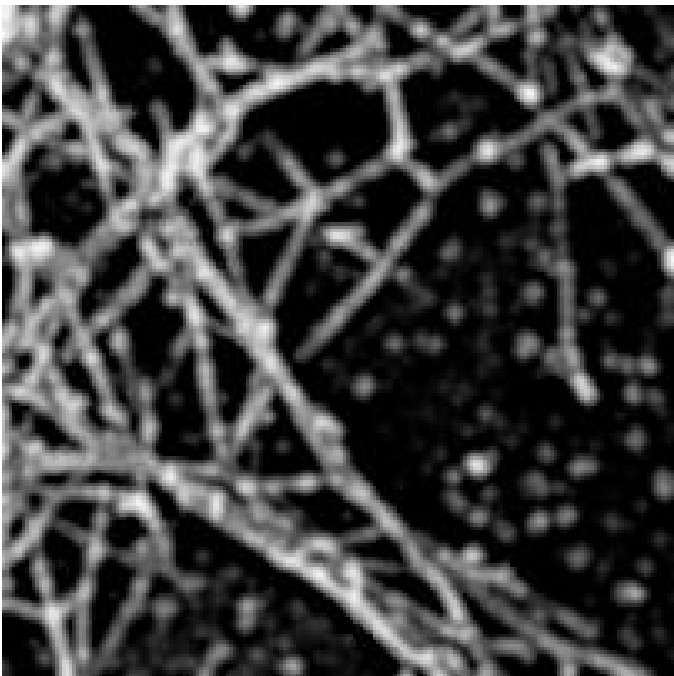

**Figure 3.** *Cont.*

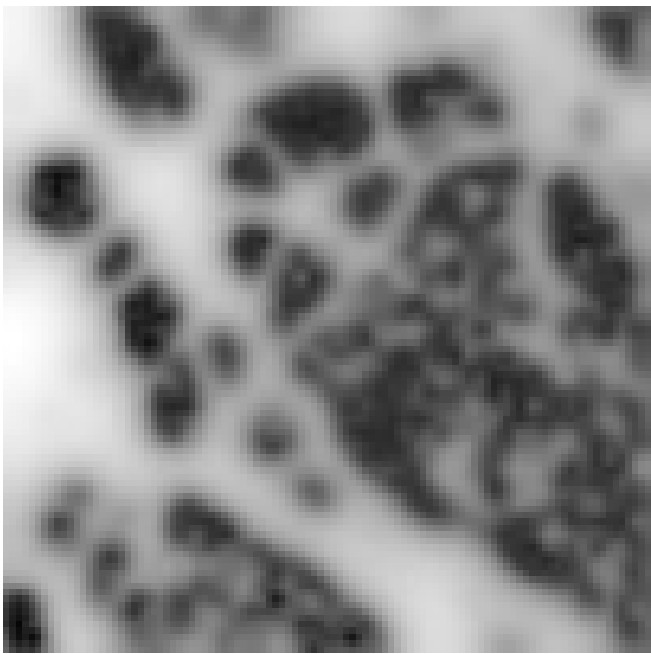

**Figure 3.** (**Up**): Portion of the cytoskeleton (Courtesy of [17]). (**Down**): reconstructed potential with the MPC scheme and $K = 5$.

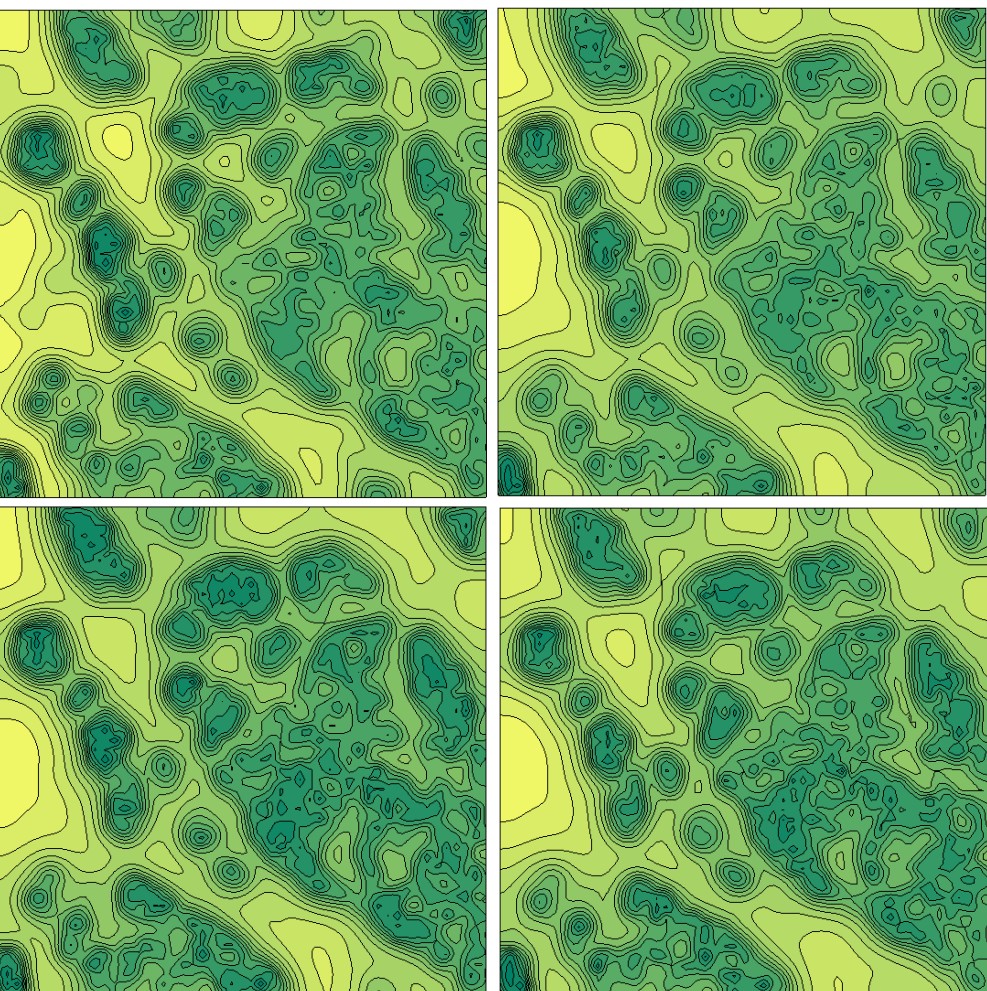

**Figure 4.** *Cont.*

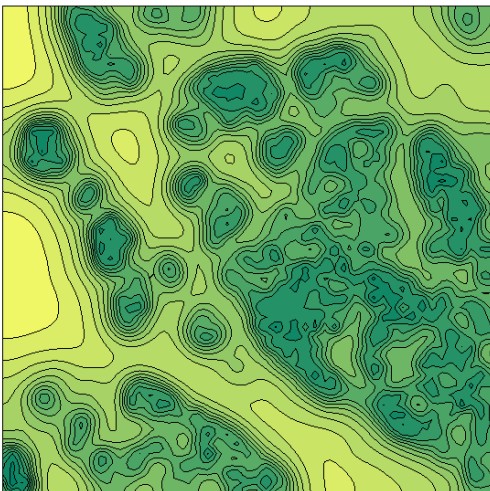

**Figure 4.** Sequence of the 5 (from **top-left** to **right-down**) calculated potentials obtained with the MPC procedure. Cross-correlation values: 0.82, 0.81, 0.82, 0.81, 0.82.

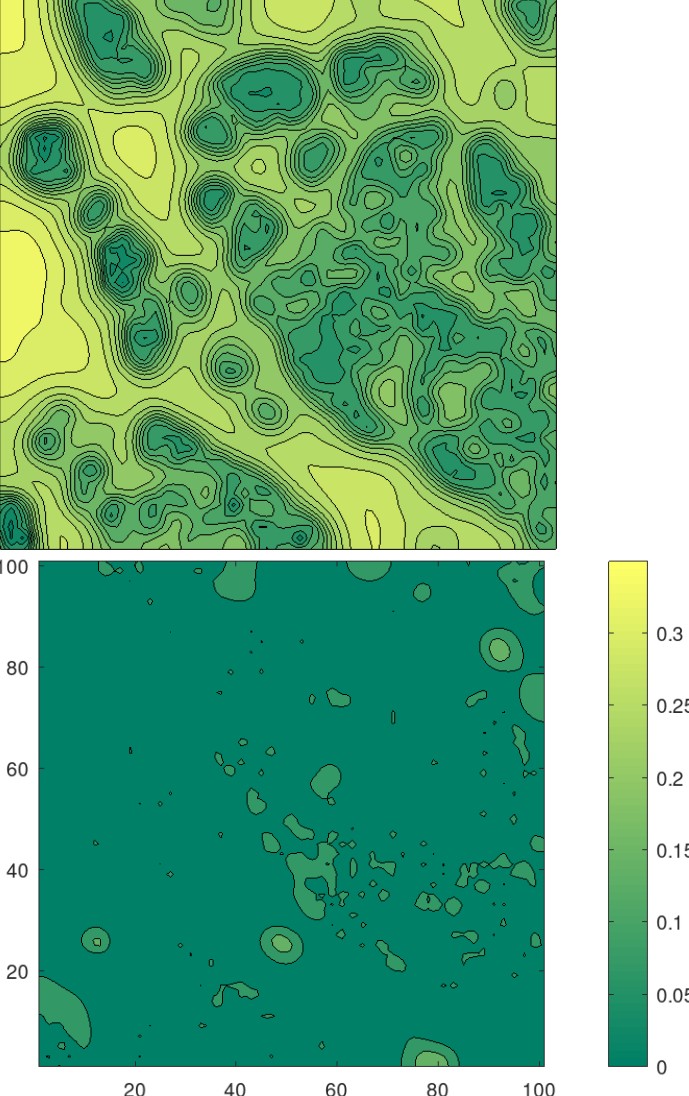

**Figure 5.** (**Up**): reconstructed mean potential. Its cross-correlation value with respect to the real image is 0.82. (**Down**): standard deviation of the reconstructed potential, in level set representation.

## 5. Resolution of FP-Based Image Reconstruction

In this section, we investigate the optical resolution of our reconstruction method, that is, try to determine a confidence value related to the scale at which our method can resolve variations of the potential. As a guideline, we remark that single molecule localization microscopy (SMLM) can distinguish distances of molecules of approximately 20 nm resolution. Therefore, we assume this resolution range of the fluorescently labeled particles images, and we attempt to quantify the smallest scale at which geometric features of the reconstructed potential $U$ can be distinguished.

For our purpose, we consider the following 'target potential', appearing as an alternating sequence of black and white circles (likewise those in test targets used for the resolution measurement of optical instruments), to synthetically generate the motion data of particles. We have

$$U(x,y) = A \left( 1 + \cos \left( \frac{2\pi}{dl} (x^2 + y^2) \right) \right), \qquad (x,y) \in \Omega, \tag{19}$$

where $A$ denotes the semi-amplitude of the variation between the minimum and the maximum of the potential, $l$ is the length of the side of the domain, $d$ is the distance between two peaks of the potential as a fraction of $l$.

Now, we consider a single MC simulation of 500 particles with the setting: $\sigma = 0.5$, $T = 90$ and $L = 3000$ frames, integrated with the time step $\tau = 0.03$. In Figure 6, we show (left) the given potential with $A = 0.05$ and $d = 1/20$, with a gray-scale value representation conveniently adjusted for illustration pourpose. According to the above working hypothesis, we suppose that the pixel's width of the image is 20 nm. In Figure 6 (left), we depict $U$ in a square of side of 500 pixels, corresponding to $\tilde{l} = 10$ μm. Hence, the distance between two peaks is $\lambda = 10/20 = 500$ nm. Further, the particle's density is 5 particles per μm$^2$, the diffusion coefficient $D \simeq 0.3472$ μm$^2$/s, and the potential depth, i.e., the difference between the maximum and the minimum, is $\tilde{U} = 0.8 \, K_B T$. For the reconstruction process, we use a binning of $50 \times 50$, $\alpha = 10^{-4}, \xi = 1$. In the numerical setting, we use a grid of $100 \times 100$ points, and $K = 5$. Also in Figure 6 (right), we show the reconstructed potential $\langle U_r \rangle$ and notice its high accuracy that is also confirmed by the high value 0.82 of the cross correlation. Notice, that the quality of the reconstruction can be further improved by using post-processing techniques of images.

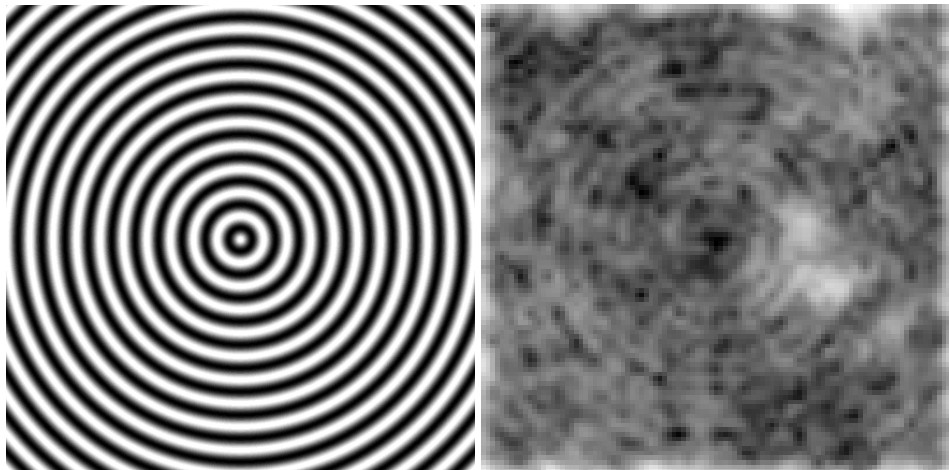

**Figure 6. (Left)**: the potential (19) with $A = 0.05$ and $d = 1/20$ (the gray scale levels spans from $U = 0$ to 0.1). **(Right)**: result of the reconstruction with the gray levels expanded to the min/max of $\langle U_r \rangle$. The cross correlation between the two images is 0.82.

Now, with the aim to define a criteria to establish the resolution measure for the potential, we introduce a confidence level for the quality of the reconstructed potential by setting a threshold for the calculated cross-correlation. This approach has been adopted in [29] for the detection of cellular objects from images acquired from electron tomography.

For that purpose, the authors used the threshold value of 0.5, whereas in our case, we set a more strict threshold-*cc* level equal to 0.8. With this threshold, we can state that the test pattern depicted in Figure 6 (left) is satisfactorily reconstructed and determines that the resolution measure associated to our 'imaging instrument' is 500 nm. Notice that this value is affected by the value of the potential *U* and the diffusion *D*, and it can be further improved by changing the other parameters of the experiment, such as *K* or the time *T* of the motion sampling.

## 6. Conclusions

A novel method for the analysis of super-resolution microscopy images was presented and applied to the reconstruction of the structure of a cell membrane potential based on the observation of the motion of particles on the membrane.

The working principle of this method is the modeling with the linear Fokker–Planck equation of the ensemble of the stochastic trajectories of particles moving on the membrane of a cell, and the solution of an optimization problem governed by this equation, where the purpose of the optimization is to find a potential such that a least-squares best fit term of the computed and observed particles' density and a Tikhonov regularization term are minimized.

Results of numerical experiments were presented that demonstrated the ability of the proposed method to reconstruct the potential of a cell membrane by using the data of a super-resolution microscopy of luminescent activated proteins.

## 7. Brief Documentation of the Code

The program code has been written in C++ with the support of Armadillo, LAPACK, and others common public available linear algebra routines. The structure of the code follows the object oriented programming style, with some basic classes for the the definition of the mathematical model, numerical grids and solvers, and derived classes for the solution of the optimization problem. Notice that the code *is not computationally optimized*, that is, it has been written with the purpose to test the ability of the numerical scheme to accomplish the proposed reconstruction task. Further major improvements could be implemented, such as a better organization of the hierarchical class structure and a more efficient use of the pointers.

The package of the source codes is composed of 14 files (The code will be available in the repository of this journal as a supplemental material):

- `CALC_FIG3.cpp, CALC_FIG6.cpp` the main driver routines
- `Non_lin_conjug_grad.cpp/.hpp` class definitions for the nonlinear conjugate gradient method
- `Optim_problem.cpp/.hpp` class definition for the optimization problem
- `BDF_Chang_Cooper.cpp/.hpp` class definitions of the BDF2 solver associated with the Chang-Cooper method
- `Chang_Cooper.cpp/.hpp` class definitions of the Chang-Cooper numerical method
- `gradients.cpp/.hpp` support functions for the discrete gradients
- `model.cpp/.hpp` class definition for the Fokker–Planck model

For our tests, the following libraries were installed on the OS Linux Mint (20.3): Armadillo (version 12.6), Openblas (version 0.3.8), Lapack (version 3.9.0), SuperLU (version 5.2.1) and HDF5® (version 1.10.4). The `makefile` gives support for the compilation, provided that the user customizes the library paths according to his own OS configuration. The compilation process is invoked with the command `make CALC_FIG3` that produces the related executable file. The command `make clean` cleans all the object codes. At run time, the executable file loads the input data from the folder `input_data`, prints on the screen some computation information about the minimizing process, and finally saves the resulting reconstruction in data files and pictures inside automated created folders.

Here follows a brief description of the input/output data structure.

The main input data of the algorithm, namely $f_d$, is a 3-dimensional matrix that, according to the syntax of Matlab®/Octave [32,33], has size as `[Nx,Ny,Nt]=size(fd)`, where `Nx,Ny` are the number of histogram bins in each dimension on the domain space, and `Nt` is the number of time step samples of the particles trajectories, i.e., `fd(:,:,n)` represents the 2-dimensional matrices slice at the time step `n` with the values of the histograms of the particles positions. This matrix is saved in the HDF5® format [27] under the set data name `value`, as follows

```
name_h5 = [save_name '.h5'];
h5create(name_h5,'/value',size(fd));
h5write(name_h5,'/value',fd);
```

The library Armadillo provides the loading method.

The files `CALC_FIG3.cpp` and `CALC_FIG6.cpp` are the driver programs for calculating the figures of the papers. They differs on the input/output data name and some parameters value. Here follows the list of variables that can be customized by the user:

- `base_data_dir`: folder name containing the input data.
- `data`: file name of the input data, i.e., the trajectories, binned in histograms, of the particles. The HDF5 format is used.
- `base_dir`: is the parent folder name where the results of the computation will be saved. This folder must be created in advance, otherwise the run-time is stopped by an error. The program creates automatically the sub-folder and saves the results inside.
- `save_name`: is a root file name, used to create file names for saving the results of the reconstruction.
- `nlcg_param_fname`: file name for the file containing the parameters of the nonlinear conjugate gradient algorithm. See below the description.
- `sig`: intensity of the Gaussian noise $\sigma$ in the FPE (4).
- `alfa`: weight $\alpha$ of the norm of the potential $U(x)$ in the objective functional (8).
- `beta`: weight of the norm of the gradient of the potential $\nabla U(x)$ in the objective functional (8). Notice, here we assume `beta`$= \alpha$.
- `xi`: weight $\zeta$ of the terminal condition in the objective functional (8).
- `ax, bx, ay, by`: are the boundaries of the 2-dimensional domain. It is not necessary to change the default values.
- `TT`: total time interval for the reconstruction. It is equal to the time of the Monte Carlo simulation or the time length of the sampled trajectories.
- `Nx`: number of grid points of the numerical domain. In this implementation it must be square.
- `mux, muy, s0`: parameters for the 2-dimensional Gauss PDF used as initial condition $f_0(x)$ for the FPE (4). `mux, muy` are the $(x, y)$ coordinates of the mean value, `s0` is the standard deviation. If `s0` $< 0$ then the PDF is uniform.
- `Nt_seq`: number of windows of the Model Predictive Control. It corresponds to the number of the reconstructions computed along all available input time step data of the particles trajectories from the Monte Carlo simulation, as depicted in Figure 4.
- `lam0`: is the initial step length for a single cycle of the NLCG algorithm.
- `kmax`: maximum number of iterations for a single cycle of the NLCG algorithm.
- `max_restart`: maximum restart sequences of the NLCG algorithm.

`Non_lin_conjug_grad.hpp/.cpp` defines the class `NLCG` that implements the nonlinear conjugate gradient method. It contains some default values of parameters for the Armijo condition during the linesearch algorithm (for details see [12,20]).

- `BACK_TRACK 0.5` is the backtracking coefficient of the linesearch algorithm.
- `MAX_LEVELS 10` is the maximum number of level search for the Armijo condition.
- `GAMMA 0.1` is the coefficient for the Armijo sufficient decrease condition of the functional.
- `EPS_CONV 1e-4` is the tolerance level for the search of the step-length

- `U_TOLERANCE 1e-4` is the tolerance level for the search of the step-length related to the control.
- `EPS_PDF 1e-8` is the minimum value of the PDF under which the control should be set vanishing (not used).

These values can be overwritten by those defined in an external file pointed out by the variable `nlcg_param_fname`. Such as an example, in the file `nlcg_params.dat` the following values are changed

- `BACK_TRACK 0.3`
- `GAMMA 0.2`

Outline of the classes usage.

The source codes `CALC_FIG3.cpp` and `CALC_FIG6.cpp` give the guidelines on how to use the classes for solving the optimization problem. After loading the input data and defining some parameters, the classes `Param` and `Model` must be instantiated. In particular, the class `Model` named `fokker_planck` that contains the mathematical model of the FPE. Afterward, it follows the definition of the Cauchy initial condition. Then it starts the iterations over the reconstruction windows related to the MPC. Each iteration solves an optimization problem, inside of that it instantiates a `Grid` class for the numerical grid and matrices for the data. The instance of the class `NLCG` provides the numerical optimizer for the reconstruction problem. The constructor of the class takes as input arguments the classes `Model` and `Grid`, and a reference to the output, i.e., the solution of the optimization stored in `u`. After loading the algorithm parameters, the optimization starts by invoking the public method `start_restart_sequence`. At the completion of the method, the solution is stored in `u`, and the final value of the PDF is set as the initial condition for the next temporal window.

Finally, notice that the classes can be easily reused, e.g., for solving the sole Fokker–Planck equation, or modified in order to implements others optimization algorithms.

**Supplementary Materials:** The following supporting information can be downloaded at: https://www.mdpi.com/article/10.3390/mca28060113/s1.

**Author Contributions:** All authors contributed equally to the formulation, analysis and the writing of the manuscript. The author M.A. made the main work in the development of the code. All authors have read and agreed to the published version of the manuscript.

**Funding:** This research received no external funding.

**Data Availability Statement:** The data presented in this study are available in Supplementary Material.

**Conflicts of Interest:** The authors declare no conflict of interest.

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
