# Peer review of "Fokker–Planck Analysis of Superresolution Microscopy Images"

_mca, doi:10.3390/mca28060113_

Round 1

Reviewer 1 Report

Comments and Suggestions for Authors

Observing molecular fluorescence remains a crucial taks in many applications in chemistery (photophysics and photochemistry).

The amont of informations has increased in the past years : it permits to improve methodologies in this field.

 In SRM microscopic image acquisitions framework, the motion of the molecules can be computed via stochastic PDE.

 The authors have an excellent background on these fields, with high quality recent works.

 In this paper, they propose to improve recent work where they made the hypothesis of interacting particles whicg leaded to important CPU time requiring a large amount of computations. In this paper, the added value is based on the fact they consider that particles do not iteract and there also consider a linear FP model. They propose a new model for  the analysis of superresolution microscopy images, they show applications to these models to the reconstruction of the structure of a cell membrane potential (based on observation of the motion of particles on the membrane).

 To this mater, as they did in previous works, they use the framework of  the Fokker-Planck equation, the modelling leads to the solution of an optimization  problem governed by this equation, where the purpose of the optimization is to find a potential such that a least-squares best fit term of the computed  and observed particles’ density and a Tikhonov regularization term are minimized.

 I also appreciate the section where the authors investigate the optical resolution of our reconstruction method, that is, try to determine a confidence value related to the scale at which our method can resolve variations of the potential.

Numerical experiments are presented: they demonstrate the efficiency of the approach in order to reconstruct the potential of a cell membrane.

 Quality of several images should be improved, and maybe including bigger images should of interest to allow the reader do better see several details.

Giving some limitations of the method (or cases where it doesn’t work, because for example sensitivity of the mode related to specific dataset) should be of interest.

 Numerical results are convincing and show the efficiency of their approach. Moreover, the discussion/documentation of the proposed C++ code (that uses Lapack packages) is really of interest.

Note that this code will be linked to this paper and accessible to the scientific community.

 This work includes an applied problems, a math modelling, and C++ code, it fits with the journal goals.

I consider this work of interest to be published in MDPI MCA.

Reviewer 2 Report

Comments and Suggestions for Authors

review of "Fokker-Planck analysis of superresolution microscopy images"

My comments below are to be read with the caveat that I am not familiar with related research on inference of microscopy images so I cannot judge novelty of the presented approach.

The article presents an interesting inversion method for the inference of a potential U underlying a particle dynamics which is indirectly measured by microscropy, but without individual particle trajectory resolution. I thought the question was very interesting and the Fokker-Planck approach definitely seems like a reasonable thing to do.

I also like that the model does not stay purely academical, but that there is an attempt to connect this concretely with realistic length scales and data.

My main issue with the manuscript is the intermediate use of 2d histograms as a surrogate for the true density f_t at time t. The authors comment on the fact (p.10, l.198) that f_d (the histogram) is very irregular, but that they do not perform smoothing of the data. I assume this is done in order to not reduce resolution any further, but I don't understand how the (not even continuous) function f_d can be used in place of f when solving the PDE (10). I am assuming this is done via a weak formulation such that the histogram is integrated over with respect to a test function, but then doesn't this in fact amount to some kind of smoothing of the histogram (via integration wrt a test function)? I might be wrong here, of course, or misunderstanding this point. Could you please clarify this?

Elaborating further, might it not be more reasonable to directly understand the microscopy measurements (i.e. the particle positions in figure 2) as random samples from the (unknown) density f, and, e.g. (there might be better ways) to
a) reconstruct f from the data via a kernel density estimator (a box kernel would be similar to the histogram, but would not be centered around the fixed grid) OR
b) to include another layer of inference, inferring f from the data. This would (I assume) add another term to the functional J(f,u) in equation (8), possibly a likelihood term (of obtaining the data for a given candidate density f).

I am just not very comfortable with histograms in general since they are very numerically and statistically unstable due to the fixed-grid binning.

Pedagogical/presentation comments:
* Maybe explain earlier that the potential U is the object of interest (rather than, say, finding a super-resolution localisation of the observed particles). This was not immediately clear to me
* Just to make sure I don't misunderstand figure 1: "White structures" in the top figure act as a repellent/barrier against the particles' motion?
* It was unclear to me whether figure 1 is to be interpreted in 2d or 3d. Of course, all physical things are 3d, but can we assume that this object is flat enough that we can ignore depth in this model? Otherwise, a particle might be observed "crossing barriers" visually, but only because the barrier does not extend fully "front-to-back".
* I am assuming the parameter \beta_n in Algorithm 2 is just the usual nonlinear CG thing, but maybe should be defined nevertheless.

Minor points:
* there are a few typos in the manuscript which I am sure the editing process will catch, but here's a sample:
    - l. 27 "stochastics"
    - l. 68 "noniteracting", as well as "and there consider"
    - l. 90 "an uniform"
    - l. 119 "of a"
    - l. 180 "sequence 2-dimensionsal"

Comments on the Quality of English Language

some minor editing needed
